# Prediction of Soft Tissue Sarcoma from Clinical Characteristics and Laboratory Data

**DOI:** 10.3390/cancers12030679

**Published:** 2020-03-13

**Authors:** Taketsugu Fujibuchi, Joji Miyawaki, Teruki Kidani, Hiroshi Imai, Hiromasa Miura

**Affiliations:** Department of Bone and Joint Surgery, Ehime University Graduate School of Medicine, Shitsukawa, Toon, Ehime 791-0295, Japan; geomiya@nifty.com (J.M.); teruteru@m.ehime-u.ac.jp (T.K.); hiroimai@m.ehime-u.ac.jp (H.I.); miura@m.ehime-u.ac.jp (H.M.)

**Keywords:** soft tissue tumor, diagnosis, clinical characteristics, laboratory data, biomarker

## Abstract

The accurate diagnosis of soft tissue tumors may be difficult. Simple clinical characteristics or laboratory data that can predict tumor malignancy can be useful tools for diagnosing soft tissue tumors. Between 2003 and 2018, 588 patients with primary soft tissue tumors were retrospectively reviewed. Their clinical characteristics and laboratory data were evaluated to determine their association with the diagnosis of benign, intermediate, or malignant tumor. Multivariable analysis revealed that tumor size ≥ 5.6 cm (odds ratio (OR), 6.15; *p* < 0.001), white blood cell (WBC) count ≥ 5700/µL (OR, 2.49; *p* = 0.002), hemoglobin (Hb) count ≤ 12.4 g/dL (OR, 2.56; *p* = 0.004), C-reactive protein (CRP) level ≥ 0.17 mg/dL (OR, 2.64; *p* < 0.001), and lactate dehydrogenase (LDH) level ≥ 240 IU/L (OR, 4.94; *p* < 0.001) were significant predictive factors for sarcoma. The sensitivity and specificity in the presence of three or more predictive factors for detecting malignant tumors were 0.58 and 0.90 respectively, and it was an appropriate threshold with the maximum Youden’s index of 0.49. Simple clinical and laboratory data were useful tools for predicting whether the tumor is malignant. Patients with soft tissue tumors that meet any three or more predictive factors should be referred to a specialist.

## 1. Introduction

There is no single clinical examination to facilitate the accurate diagnosis of soft tissue sarcomas, which requires comprehensive consideration of several examinations. In addition to their complexity, soft tissue tumors belong to a heterogeneous group of relatively rare tumors; moreover, malignant soft tissue tumors, i.e., soft tissue sarcomas are exclusively rare [1]. Several physicians, with the exception of specialists, do not encounter them often in daily practice. Thus, they are not familiar with the process of diagnosing soft tissue tumors. Even specialists may occasionally find the differentiation into benign and malignant tumors challenging. Therefore, soft tissue sarcomas are sometimes misdiagnosed as benign lesions, which can delay the correct diagnosis or result in inadequate treatment such as unplanned resections with poor prognosis [2]. Several studies have focused on determining the clinical characteristics that correlated the prognosis of soft tissue sarcoma. They reported that age, duration of symptoms, tumor size [3], serum C-reactive protein (CRP) level [4], and comorbidity [5] are prognostic factors for soft tissue sarcomas. Several groups, including the Union for International Cancer Control [6] and the American Joint Committee on Cancer [7], have used tumor size and depth of location as the standard criteria for staging soft tissue sarcomas. However, there are few reports on the factors associated with the diagnosis of soft tissue sarcoma. Clinical histories and findings from physical examination are sometimes useful in the diagnostic process of soft tissue tumors. Moreover, imaging modalities, especially magnetic resonance imaging (MRI), are powerful tools for the diagnosis of soft tissue tumor. However, several tumors present with nonspecific findings. Furthermore, patients with soft tissue sarcomas do not always initially consult a bone and soft tissue tumor specialist and usually see a primary care physician first. It is important that primary care physicians should be able to distinguish soft tissue sarcomas from benign tumors and decide whether referral to specialist is needed or not. Simple clinical data that can predict the benignity or malignancy of given tumor can be useful for not only specialists, but also primary care physicians, while diagnosing soft tissue tumors. Avoiding misdiagnosis can improve the prognosis in such patients.

The aim of this study was to clarify the relationship between the diagnosis of benign or malignant tumor and simple clinical characteristics or laboratory data: the patient’s age, sex, tumor size, lesion depth, white blood cell (WBC) count, hemoglobin (Hb), serum C-reactive protein (CRP) level, and serum lactate dehydrogenase (LDH) level. We also aimed to explore the possibility of predicting the benign or malignant nature of the tumor from simple clinical characteristics and laboratory data.

## 2. Results

### 2.1. Patient Demographics

Of the 588 patients that fulfilled the selection criteria, 457 patients had benign soft tissue tumors, 40 had intermediate tumors, and 91 patients had malignant soft tissue tumors. The study population included 289 men and 299 women, and 203 tumors were superficial, while 385 were deep. The median age of the patients was 56 years (range, 2–87 years) in benign tumor group, 69 years (11–83 years) in intermediate tumor group, and 65 years (2–97 years) in malignant tumor group. The median tumor size, WBC count, Hb count, serum CRP level, and serum LDH level were 4.2 cm (0–27.3 cm), 5500/µL (2000–11,600/µL), 14.1 g/dL (7.7–18.1 g/dL), 0.05 mg/dL (0.01–8.03 mg/dL), and 180 IU/L (65–441 IU/L) in the benign tumor group; 11.5 cm (2.6–33.0 cm), 5800/µL (3900–12,200/µL), 14.2 g/dL (10.7–17.2 g/dL), 0.07 mg/dL (0.01–0.64 mg/dL), and 176 IU/L (129–294 IU/L) in the intermediate tumor group; and 8.8 cm (1.0–31.6 cm), 6500/µL (2500–18,100/µL), 13.2 g/dL (6.3–17.4 g/dL), 0.19 mg/dL (0.01–21.2 mg/dL), and 182 IU/L (117–2555 IU/L) in the malignant tumor group, respectively. Lipoma was the most common diagnosis, followed by schwannoma and hemangioma in the benign-tumor group; atypical lipomatous tumor/well differentiated liposarcoma (ALT/WDL), followed by desmoid tumor in the intermediate-tumor group; and undifferentiated pleomorphic sarcoma (UPS) including malignant fibrous histiocytoma (MFH) and followed by leiomyosarcoma and myxofibrosarcoma in the malignant-tumor group (Table 1). The thigh was the most common location among all the groups, followed by the hand or finger in the benign-tumor group; chest wall/back in the intermediate-tumor group; and chest wall/back or leg in the malignant-tumor group (Table 2). 

### 2.2. Determination of Optimal Thresholds of Clinical Characteristics and Laboratory Data

Optimal thresholds were determined using receiver operative characteristic (ROC) curve and Youden’s index as shown in Table 3. According to these results, patients were classified into two categories by age: ≥ 72 years and < 72 years. Maximal tumor size was divided into two categories: ≥ 5.6 cm and < 5.6 cm. Laboratory parameters were stratified as follows: WBC, ≥ 5700/µL versus < 5700/µL; Hb, > 12.4 g/dL versus ≤ 12.4 g/dL; CRP, ≥ 0.17 mg/dL versus < 0.17 mg/dL; and LDH, ≥ 240 IU/L versus < 240 IU/L.

### 2.3. Association between the Diagnosis and Clinical Characteristics or Laboratory Data

Univariate analysis of the diagnosis of soft tissue sarcoma and clinical characteristics or laboratory data revealed a significant association between diagnosis and age (*p* < 0.001), tumor size (*p* < 0.001), WBC count (*p* < 0.001), Hb count (*p* < 0.001), CRP level (*p* < 0.001), and LDH level (*p* < 0.001). Older patients with large tumors, high WBC count, low Hb count, high serum CRP level, and high serum LDH level were likely to be diagnosed with soft tissue sarcoma (Table 4). 

### 2.4. Predictive Factors for Soft Tissue Sarcoma on Multivariate Analysis

Multivariable analysis revealed that large tumor size (OR, 6.15; 95% confidence interval (CI), 3.34–12.0; *p* < 0.001), high WBC count (OR, 2.49; 95% CI, 1.40–4.57; *p* = 0.002), low Hb count (OR, 2.56; 95% CI, 1.36–4.77; *p* = 0.004), high CRP level (OR, 2.64; 95% CI, 1.51–4.59; *p* < 0.001), and high LDH level (OR, 4.94; 95% CI, 2.40–10.3; *p* < 0.001) were significant predictive factors for soft tissue sarcoma (Table 5). 

### 2.5. Association between Histological Grade or Metastatic Status and Clinical Characteristics or Laboratory Data 

Significant association was observed between the histological grade and serum CRP level (*p* = 0.005). Patients with distant metastases at the first visit showed an association with large tumor size (*p* = 0.036) and high serum LDH level (*p* < 0.001) (Table 6).

### 2.6. Sensitivity and Specificity According to the Applicable Number of Predictive Factors

The sensitivity and specificity for predicting soft tissue sarcoma according to the applicable number of predictive factors are shown in Table 7. The area under the curve (AUC) of the ROC curve was 0.83. The sensitivity for detecting soft tissue sarcoma was 0.58 in the presence of three or more predictive factors and specificity for detecting soft tissue sarcoma was 0.90, and it was an appropriate threshold with the maximum Youden’s index of 0.49 (Table 7, Figure 1).

### 2.7. Sensitivity Using Threshold by Pathological Diagnosis

Number of patients that meet the threshold by pathological diagnosis was evaluated in common sarcomas (Table 8). Sensitivity varies according to morphology of sarcomas. Leiomyosarcoma showed the highest sensitivity of 0.73, followed by pleomorphic liposarcoma, whereas myxoid liposarcoma showed the lowest sensitivity of 0.25.

## 3. Discussion

In this study, large tumor size, high WBC count, low Hb count, high CRP level, and a high LDH level were shown to be significant predictive factors for malignant tumors, and the presence of three or more predictive factors was an appropriate threshold with the maximum Youden’s index.

The diagnosis of soft tissue tumors is based on clinical, laboratory, imaging, and pathological findings. This approach is complex and requires a certain level of sophistication and expertise. Therefore, soft tissue sarcomas are sometimes misdiagnosed, resulting in unplanned resections or diagnostic delays. Although reports on the relationship between the additional wide excision after unplanned excision and local recurrence of soft tissue sarcomas are controversial [8,9,10,11], unplanned excision is generally considered to be associated with an increased risk of local recurrence and the needed for additional and more extensive surgery [2]. A delay in diagnosing a sarcoma will inevitably lead to tumor enlargement and large tumor size is a very poor prognostic factor [12]. Successful prediction of soft tissue sarcoma using simple clinical characteristics and laboratory data will help in preventing misdiagnosis and improve the patient’s prognosis.

This study found no significant relationship between age and diagnosis using multivariate analysis. However, the significance of age varies for different studies [13,14,15]. Deep-seated tumors were reported to have a higher risk of malignancy by a majority of studies [2,16]. On the other hand, one study reported that the lesion depth (with respect to the superficial investing fascia) was less important as a predictor of malignant potential [15]. Our findings also showed that lesion depth was not a predictive factor for soft tissue sarcoma, and conversely, benign tumors tended to be associated with deep location (according to the analysis). This may be attributed to the fact that many patients in this study were referral patients and several deep-seated tumors were suspected to be soft tissue sarcomas. Our study supported the findings of other studies on tumor size. Thresholds vary for different studies, and a tumor size ≥ 5.6 cm was a predictive factor for sarcoma in this study. Few studies have examined the association between clinical characteristics and diagnosis of soft tissue tumors (Table 9) [13,14,15,17]. However, studies on the association between laboratory data and soft tissue tumor diagnosis are lacking.

Previous studies reported that high WBC count, low Hb count, high serum CRP level, and high serum LDH level were factors associated with poor prognosis [4,18,19,20,21]. We confirmed that high WBC count, low Hb count, high serum CRP level, and high serum LDH level were independent predictive factors for soft tissue sarcoma using multivariate analysis. Coussens et al. stated that inflammation is a critical component of tumor progression [22]. Inflammation may be responsible for the high WBC count. Low Hb count, i.e., anemia, may be a result of inflammation-induced iron misutilization [21]. Additionally, an association between serum CRP level and tumor diagnosis in this study can be explained by the state of inflammation in the soft tissue sarcoma. The association between LDH levels and a diagnosis of malignancy may be explained by the Warburg effect. In normal cells, glucose is metabolized by glycolysis in a multistep set of reactions resulting in the creation of pyruvate. A large amount of pyruvate enters the mitochondria, where it is oxidized by the ATP cycle in the presence of oxygen, but pyruvate is converted into lactate by LDH only in the absence of oxygen. However, most of the pyruvate is converted into lactate even in the presence of oxygen in cancer cells. This phenomenon is known as the Warburg effect [23,24].

We arrive at the feasibility of using these results to predict whether a tumor is benign or malignant, based on the clinical characteristics and laboratory data. The value of AUC of the ROC curve was considered excellent when investigating the ability to predict malignancy based on the number of applicable predictive factors. Thus, the possibility of soft tissue sarcoma increases with an increase in the number of predictive factors in a patient with soft tissue tumor. Given a threshold, if a patient with soft tissue tumor meets any three or more predictive factors, the tumor is likely to be malignant and the patient should be referred to a specialist. Actually, sensitivity varies according to morphology of sarcomas. In this study, UPS/MFH, Leiomyosarcoma, and pleomorphic liposarcoma showed relatively high sensitivity, whereas myxofibrosarcoma, MPNST, and myxoid liposarcoma showed relatively low sensitivity.

This study had several potential limitations. First, we excluded patients with local recurrence, infection and metastatic soft tissue tumor. In fact, differential diagnoses often overlap in routine medical practice. However, the exclusion of such patients was probably appropriate for elucidating and simplifying the relationship between the diagnosis and the clinical characteristics or laboratory data. Second, many patients in this study were referral patients, so the ratio of malignant tumor to benign tumor differed from that for nonspecialized hospitals. Even if the ratio of each group are different, overall tendency of clinical characteristics or laboratory data of each group would not be different. So, the results in this study are considered appropriate. Third, we have not included tumor-specific molecules, genetic evaluation of tumor, and prognostic evaluation in analysis items. Certainly, these items were very important. However, the aim of this study was to predict soft tissue sarcoma using simple clinical characteristics and laboratory data and this aim was able to be achieved without those items.

## 4. Materials and Methods

This retrospective, case-control study reviewed 654 patients with soft tissue tumors, who were treated at our institution between April 2003 and March 2018. The study did not consider surgical treatment. Patients with recurrent tumors or residual tumors after unplanned resection (*n* = 44), those with metastatic soft tissue tumor (*n* = 9), those with other malignant tumors or inflammatory diseases (*n* = 3), and patients with incomplete clinical data (*n* = 10) were excluded. Altogether, 588 patients were included in the analysis. Pathological diagnosis including benign, intermediate, or malignant tumor; clinical characteristics including age, sex, tumor size, and lesion depth; and laboratory data including WBC count, Hb count, serum CRP level, and serum LDH level were reviewed for each patient. Pathological diagnosis was confirmed using core needle, incisional, or excisional biopsy. Not only pathological diagnosis, but also histological grade according to Fédération Nationale des Centres de Lutte Contre le Cancer (FNCLCC) grade was determined. Tumor size was defined using the maximal diameter of the tumor mass measured by MRI. Lesion depth was categorized using MRI, and tumors were classified into superficial or deep groups. Superficial tumors were defined as those located superficially to the fascia, while deep tumors were defined as those located deeply to the fascia. 

### Statistical Analysis

Optimal thresholds of each parameter were determined using ROC curve and Youden’s index. Continuous parameters of clinical characteristics and laboratory data were categorized according to the threshold and patients were classified into two groups.

Fisher’s exact test was used to analyze the association between diagnosis and clinical characteristics or laboratory data to identify the predictive factors for soft tissue sarcoma. Benign and intermediate soft tissue tumors were grouped into one category and the diagnosis was categorized into soft tissue sarcoma or others. Significant variables identified in univariate analysis and variables that were reported to be prognostic factors in the past were simultaneously evaluated using multivariable analysis. Multivariate analysis was performed using logistic regression analysis to calculate the OR and 95% CI. Analyses of the association between histologic grade and clinical characteristics or laboratory data, and metastatic status and clinical characteristics or laboratory data were also performed. In the analysis of histologic grade, FNCLCC grade 1 was classified as low grade and grade 2 and 3 were classified into high grade. A two-sided *p*-value of < 0.05 was considered significant for all statistical analyses. The relationship between the number of applicable predictive factors and diagnostic accuracy was analyzed using the AUC of ROC curve. The optimal threshold for predicting soft tissue sarcoma was determined using Youden’s index. Statistical analysis was performed using JMP^®^ 11 (SAS Institute Inc., Cary, NC, USA).

The study was conducted in accordance with the ethical standards described in the Declaration of Helsinki and was reviewed and approved by the Institutional Review Board, Ehime University Hospital (No. 1510010). Written informed consent was waived because of the retrospective nature of this study.

## 5. Conclusions

The following five simple clinical characteristics or laboratory data: large tumor size (≥ 5.6 cm), high WBC count (≥ 5700/µL), low Hb count (≤ 12.4 g/dL), high CRP level (≥ 0.17 mg/dL), and high LDH level (≥ 240 IU/L) can be useful for predicting malignancy of soft tissue tumors. A patient with a soft tissue tumor that meets any three or more predictive factors should be referred to a specialist since the tumor is likely to be a soft tissue sarcoma.

## Figures and Tables

**Figure 1 cancers-12-00679-f001:**
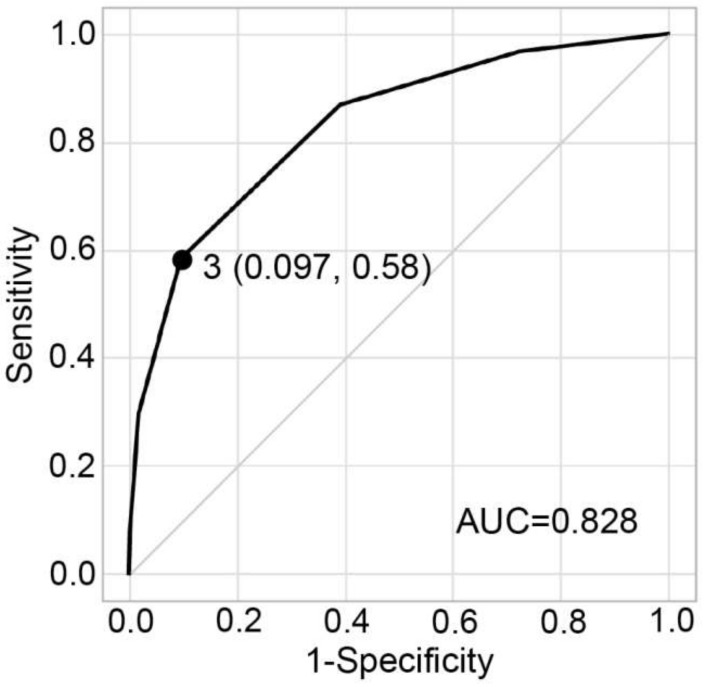
ROC analysis for the appropriate threshold of applicable number of predictive factors. The presence of three or more predictive factors was an appropriate threshold with Youden’s index of 0.49, corresponding to a sensitivity of 0.58 and 1-specificity of 0.097. ROC: receiver operative characteristic.

**Table 1 cancers-12-00679-t001:** Pathologic diagnosis of the 588 patients.

Benign Tumor	Number of Patients	Intermediate Tumor	Number of Patients	Malignant Tumor	Number of Patients
Lipoma	166	ALT/WDL	29	UPS/MFH	20
Schwannoma	110	Desmoid tumor	6	Leiomyosarcoma	11
Hemangioma	40	SFT	3	Myxofibrosarcoma	10
Tenosynovial GCT, localized type	37	Others	2	MPNST	8
Fibroma	26			Myxoid liposarcoma	8
Neurofibroma	19			Pleomorphic liposarcoma	7
Tenosynovial GCT, diffuse type	9			Dedifferentiated liposarcoma	5
Others	50			Rhabdomyosarcoma	5
				Others	17
Total	457	Total	40	Total	91

GCT: giant cell tumor, SFT: solitary fibrous tumor, ALT/WDL: atypical lipomatous tumor/well-differentiated liposarcoma, UPS: undifferentiated pleomorphic sarcoma, MFH: malignant fibrous histiocytoma, MPNST: malignant peripheral nerve sheath tumor.

**Table 2 cancers-12-00679-t002:** Location of tumors.

Location	Number of Patients			
	Benign Tumor	Intermediate Tumor	Malignant Tumor	Total
Thigh	78	22	47	147
Chest wall, back	57	7	8	72
Hand, finger	58	0	2	60
Leg	40	2	8	50
Upper arm	41	1	4	46
Forearm	35	2	3	40
Foot, toe	30	1	2	33
Shoulder	25	0	2	27
Others	93	5	15	113

**Table 3 cancers-12-00679-t003:** AUC, optimal threshold, and Youden’s index for each item.

Clinical Characteristics/ Laboratory Data	AUC of ROC Curve	Threshold	Youden’s Index
Age	0.64	≥ 72 years >	0.23
Tumor size	0.73	≥ 5.6 cm >	0.45
WBC	0.68	≥ 5700/µL >	0.30
Hb	0.66	> 12.4 g/dL ≥	0.24
CRP	0.71	≥ 0.17 mg/dL >	0.37
LDH	0.55	≥ 240 IU/L >	0.22

WBC: white blood cell, Hb: hemoglobin, CRP: C-reactive protein, LDH: lactate dehydrogenase, ROC: receiver operative characteristic, AUC: area under the curve.

**Table 4 cancers-12-00679-t004:** The association between diagnosis and clinical characteristics or laboratory data.

Clinical Characteristics/ Laboratory Data		Soft Tissue Sarcoma (Patients)	Others; Benign and Intermediate Tumor (Patients)	*p* (Fisher’s Exact Test)
Age	≥ 72 years	36	83	
	< 72 years	55	414	< 0.001
Sex	Male	53	236	
	Female	38	261	0.068
Lesion depth	Deep	57	327	
	Superficial	34	170	0.55
Tumor size	≥ 5.6 cm	75	193	
	< 5.6 cm	16	304	< 0.001
WBC	≥ 5700/µL	70	234	
	< 5700/µL	21	263	< 0.001
Hb	> 12.4 g/dL	57	429	
	≤ 12.4 g/dL	34	68	< 0.001
CRP	≥ 0.17 mg/dL	49	86	
	< 0.17 mg/dL	42	411	< 0.001
LDH	≥ 240 IU/L	26	32	
	< 240 IU/L	65	465	< 0.001

WBC: white blood cell, Hb: hemoglobin, CRP: C-reactive protein, LDH: lactate dehydrogenase.

**Table 5 cancers-12-00679-t005:** Multivariate analysis for predicting factor for soft tissue sarcoma.

Clinical Characteristics/Laboratory Data		OR	95%CI	*p*
Age	≥ 72 years	1.53	0.84–2.73	0.16
	< 72 years	1		
Lesion depth	Deep	0.76	0.44–1.32	0.33
	Superficial	1		
Tumor size	≥ 5.6 cm	6.15	3.34–12.0	< 0.001
	< 5.6 cm	1		
WBC	≥ 5700/µL	2.49	1.40–4.57	0.002
	< 5700/µL	1		
Hb	≤ 12.4 g/dL	2.56	1.36–4.77	0.004
	> 12.4 g/dL	1		
CRP	≥ 0.17 mg/dL	2.64	1.51–4.59	< 0.001
	< 0.17 mg/dL	1		
LDH	≥ 240 IU/L	4.94	2.40–10.3	< 0.001
	< 240 IU/L	1		

WBC: white blood cell, Hb: hemoglobin, CRP: C-reactive protein, LDH: lactate dehydrogenase, OR: odds ratio, CI: confidence interval.

**Table 6 cancers-12-00679-t006:** The association between histological grade or metastatic status and clinical characteristics or laboratory data.

Clinical Characteristics/Laboratory Data	Histological Grade Low Grade Versus High Grade *p* (Fisher’s Exact Test)	Metastatic Status M0 Versus M1 *p* (Fisher’s Exact Test)
Age ≥ 72 years versus < 72 years	0.069	0.054
Sex Male versus female	0.55	0.60
Lesion depth Deep versus superficial	0.36	0.27
Tumor size ≥ 5.6 cm versus < 5.6 cm	0.23	0.036
WBC ≥ 5700/µL versus < 5700/µL	0.07	0.34
Hb > 12.4 g/dL versus ≤ 12.4 g/dL	0.76	1.00
CRP ≥ 0.17 mg/dL versus < 0.17 mg/dL	0.005	0.78
LDH ≥ 240 IU/L versus < 240 IU/L	0.33	< 0.001

WBC: white blood cell, Hb: hemoglobin, CRP: C-reactive protein, LDH: lactate dehydrogenase.

**Table 7 cancers-12-00679-t007:** Sensitivity and specificity according to the applicable number of predictive factors.

Applicable Number of Predictive Factors	Malignant Tumor (Patients)	Benign and Intermediate Tumor (Patients)	Sensitivity	Specificity	Youden Index
All five factors applicable	7	1			
Others	84	496	0.077	1.00	0.075
Four or more factors applicable	27	9			
Others	64	488	0.30	0.98	0.28
Three or more factors	53	48			
Others	38	449	0.58	0.90	0.49
Two or more factors	79	195			
Others	12	302	0.87	0.61	0.48
One or more factors	88	360			
Others	3	137	0.97	0.28	0.24

Predictive factors: Maximal tumor size ≥ 5.6 cm, WBC ≥ 5700/µL, Hb ≤ 12.4 g/dL, CRP ≥ 0.17 mg/dL, and LDH ≥ 240 IU/L.

**Table 8 cancers-12-00679-t008:** Number of patients that meet the threshold by pathological diagnosis.

Pathological Diagnosis	Number of Patients	Number of Patients That Meet the Threshold	Sensitivity
UPS/MFH	20	13	0.65
Leiomyosarcoma	11	8	0.73
Myxofibrosarcoma	10	4	0.40
MPNST	8	3	0.38
Myxoid liposarcoma	8	2	0.25
Pleomorphic liposarcoma	7	5	0.71

UPS: undifferentiated pleomorphic sarcoma, MFH: malignant fibrous histiocytoma, MPNST: malignant peripheral nerve sheath tumor.

**Table 9 cancers-12-00679-t009:** Literature review of the predictive factors for sarcoma.

Author	Year	No. of Patients	Predictive Factors					
			Tumor Size (≥ 5 cm)	Location (Deep)	Age	Pain (+)	Increasing in Size	Other Predictive Factors
Myhre-Jensen [13]	1981	1403	○	○	×	NA	NA	Tumors obviously malignant for some reasons (invasion of nerves, vessels or bone)
Persson et al. [14]	1986	280	○	○	○ (≥ 50 years)	NA	NA	Tumors located proximally in the extremities
Johnson et al. [17]	2001	526	○	○	NA	○	○	
Datir et al. [15]	2008	571	○	×	○ (Increasing patient age)	NA	NA	
Fujibuchi et al.	Current study	588	(≥ 5.6 cm)	×	×	NA	NA	WBC ≥ 5700/µL, Hb ≤ 12.4 g/dL, CRP ≥ 0.17 mg/dL, LDH ≥ 240 IU/L

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
