# Peer review of "Prediction of Soft Tissue Sarcoma from Clinical Characteristics and Laboratory Data"

_cancers, 2020, doi:10.3390/cancers12030679_

Round 1

Reviewer 1 Report

I will suggest the author should also provide the range of WBC counts, hemoglobin counts and LDH release  etc. (in the table 3, 4 and 5) of a normal person just for a reference and to compare with the abnormal or patient counts.

Reviewer 2 Report

The new research article by Fujibuchi et al. is considered to be a meaningful data for the evaluation of prognostic factors for soft tissue tumors.

However, general clinical testing such as blood count and clinical chemical items; LDH, and CRP are extremely poorly specific factors, and general inflammatory markers, serum deviating enzymes, and blood cell counts vary widely in patients with malignant tumors. If new factors are being evaluated as predictors of sarcoma prognosis, tumor-specific molecules should also be evaluated.

The authors list patient profile, kinds of tumor, tumor grade (high and low), tumor size and location, and general laboratory parameters. More importantly, tumor-specific factors such as tumor type, differentiation, and genetic mutation or gene translocation must be included in the tumor assessment.

The authors should add the pathological evaluation, immunohistochemistry, chromosomal abnormalities, and genetic mutations as factors in the tumor character and prognostic evaluation. The morphological and genetic evaluation of tumor is extremely important as a prognostic predictor with tumor classification, treatment selection, and treatment response.

Grade classification for soft tissue tumors is not sufficient as an international classification unless grade is evaluated in incorporating the Fédération Nationale des Centres de Lutte Contre Le Cancer system (FNCLCC system).

Reviewer 3 Report

The author mentioned combination of clinical characteristics or laboratory data can predict tumor malignancy according to the previous reported threshold or the standard values used at their institution. They concluded good prediction that a patient with a soft tissue tumor that meets any two or more predictive factors suggests a soft tissue sarcoma and recommend to refer an oncology specialist.

How about using new threshold by statistical analysis of your data, not using previous reported data (5cm) or standard values in your institution (WBC 9100 etc). Can you get better diagnostic tool?

The diagnosis of 588 patients are 457 benign tumors and 131 intermediate to malignant tumors. This ratio is standard as cancer center but not typical for non-specialized hospitals. Malignant tumor is far rare. That implies statistical results from 588 patients of this study and your promotion might not appropriate for every hospital.

Your result that the presence of two or more predictive factors was an appropriate threshold indicates that if a geriatric patient has 5cm tumor, it would be malignant. It is not very new information and need more interesting scenario. How about a 50-year-old patient have 5 cm lipoma?

Round 2

Reviewer 2 Report

Response 2 and 3 lack the morphological assessment that is important for the evaluation of sarcomas. Since this point cannot be resolved, the answers and authors’ revised version are insufficient.

Their responses did not satisfactorily address the Reviewer comment. 

Author Response

Response to reviewer 2 comments

Response 2 and 3 lack the morphological assessment that is important for the evaluation of sarcomas. Since this point cannot be resolved, the answers and authors’ revised version are insufficient.

Thank you very much for careful reading our manuscript and for giving useful comments.
As you state, we also think morphologic analysis is important. For additional analysis, we selected some common pathological diagnoses, and examined sensitivity by pathological diagnosis. This analysis revealed that sensitivity varies according to morphology of sarcomas, and UPS/MFH, Leiomyosarcoma, and pleomorphic liposarcoma showed relatively high sensitivity, whereas myxofibrosarcoma, MPNST, and myxoid liposarcoma showed relatively low sensitivity. We have gained one more insight. Thank you very much.

Reviewer 3 Report

good revision

Round 3

Reviewer 2 Report

The manuscript was improved.

Manuscript is fine for publication.
